# Efficacy of aspirin, clopidogrel, and ticlopidine in stroke prevention: A population-based case-cohort study in Taiwan

Yi-Sin Wong[1], Ching-Fang Tsai [ID][2], Yueh-Han Hsu[2], Cheung-Ter Ong [ID][3]*

1 Department of Family Medicine, Ditmanson Medical Foundation Chiayi Christian Hospital, Chiayi City, Taiwan, 2 Department of Medical Research, Ditmanson Medical Foundation Chiayi Christian Hospital, Chiayi City, Taiwan, 3 Department of Neurology, Ditmanson Medical Foundation Chiayi Christian Hospital, Chiayi City, Taiwan

* ctong98@yahoo.com.tw

## Abstract

### Background

In real-world practice settings, there is insufficient evidence on the efficacy of antiplatelet drugs, including clopidogrel, aspirin, and ticlopidine, in stroke prevention.

### Purpose

To compare the efficacies between aspirin and clopidogrel and aspirin and ticlopidine in stroke prevention.

### Methods

This population-based case-cohort study utilized the data obtained from a randomized sample of one million subjects in the Taiwan National Health Insurance Research Database. Patients who were hospitalized owing to the primary diagnosis of ischemic stroke from January 1, 2000 to December 31, 2010 and treated with aspirin, ticlopidine, or clopidogrel were included in the study. Propensity score matching with a 1:4 ratio was performed to compare aspirin with ticlopidine and clopidogrel. The criteria for inclusion were the use of one of the three antiplatelet drugs for more than 14 days within the first month after the stroke and then continued use of the antiplatelet drugs until the study endpoint of recurrent stroke.

### Results

During the 3-year follow-up period, the recurrent stroke rates were 1.62% (42/2585), 1.48% (3/203), and 2.55% (8/314) in the aspirin, ticlopidine, and clopidogrel groups, respectively. Compared with the patients treated with aspirin, those treated with clopidogrel and ticlopidine showed competing risk-adjusted hazard ratios of recurrent stroke of 2.27 (1.02–5.07) and 0.62 (0.08–4.86), respectively.

**Data Availability Statement:** This retrospective study used data from the Taiwan National Health Insurance Research Database (NHIRD). The NHI Research Institute provides the database to

researchers after anonymizing all personal information. Researchers may only share the data resulting from their analysis; in the study, all of this data is within the manuscript and the Supporting Information. Researchers who would like to access the full data underlying this study can apply to the Data Science Centre of the Ministry of Health and Welfare (MOHW) of Taiwan (https://dep.mohw.gov.tw/dos/np-2497-113.html). To apply, the researcher must design a research plan and gain IRB approval, after which point the research plan can be submitted to the MOHW for approval. The data in the manuscript were retrieved from LHID 2005. The IRB number of the study is CYCH-IRB 2018013. The search code were ICD-9 code (433, 434 and 436) for ischemic stroke. Medication search code were aspirin, clopidogrel and ticlopidine. The search for risk factor of stroke and complication or adverse effect also use ICD-9 code.

**Funding:** The author(s) received no specific funding for this work.

**Competing interests:** The authors have declared that no competing interests exist.

## Conclusion

Compared with the patients treated with aspirin, those treated with clopidogrel were at a higher risk of recurrent stroke. For stroke prevention, aspirin was superior to clopidogrel whereas ticlopidine was not inferior to aspirin.

## Introduction

In Taiwan, approximately 150,000 individuals experience first ever ischemic strokes every year [1]. Stroke is a major cause of mortality and morbidity and constitutes a high risk of recurrent stroke. The annual recurrent ischemic stroke rate in Taiwan is approximately 10%, which is higher than that reported in the United States [2]. Over the past four decades, stroke incidence has decreased significantly in high-income countries; however, a similar trend has not been observed in low-income countries. This observed reduction in stroke incidence in specific countries is most likely related to implementing preventive therapies and effective control of stroke risk factors [3]. Besides controlling blood pressure and sugar, and the treatment of hyperlipidemia, antiplatelet agents for non-cardioembolic stroke are the most important factors for preventing recurrent stroke in patients with stroke or transient ischemic attack. At present, the US Food and Drug Administration has approved four antiplatelet treatments to prevent ischemic stroke: aspirin, aspirin in combination with dipyridamole, clopidogrel, and ticlopidine [4].

Aspirin's antithrombotic effect is due to the irreversible acetylation of platelet cyclooxygenase-1 and inhibition of thromboxane A2 synthesis [5]. Adenosine diphosphate (ADP) is a platelet activator that is released from red blood cells. It activates platelets to induce platelet adhesion and aggregation. Clopidogrel and ticlopidine inhibit platelet aggregation by inhibiting adenosine diphosphate [6]. Further, like aspirin, both ticlopidine and clopidogrel prevent thrombosis and related cardiovascular and cerebrovascular events [6–10]. Aspirin is the most commonly prescribed antiplatelet drug for stroke prevention and can significantly decrease the recurrent stroke rate. However, the annual recurrent ischemic stroke rate remains above 3.58% [11, 12]. A study comparing the efficacy of ticlopidine with aspirin for the prevention of recurrent stroke in non-white patients found that ticlopidine was superior to aspirin [12]. In contrast, Gorelick et al. investigated the efficacy of aspirin and ticlopidine for recurrent stroke prevention in African American patients showed no significant difference in the recurrent stroke rate between the aspirin and ticlopidine groups [8]. Conversely, the Clopidogrel versus Aspirin in Patients at Risk of Ischemic Events (CAPRIE) trial demonstrated that clopidogrel was more effective in preventing ischemic stroke, myocardial infarction, and death related to vascular disease than aspirin [13].

Given the relatively high incidence of recurrent stroke, several studies investigated potential preventive treatment approaches in patients who failed stroke prevention by aspirin. They did this by testing combination treatments, including aspirin plus clopidogrel and clopidogrel plus ticlopidine [8, 14, 15]. These studies revealed that all three antiplatelet drugs reduced the risk of recurrent stroke. Recurrent stroke in patients on antiplatelet drugs for stroke prevention is a common concern. Studies previously showed that switching from aspirin to another antiplatelet drug or combining aspirin with another antiplatelet agent was associated with improved prevention of recurrent stroke compared with the maintained use of aspirin alone [1, 11].

Recent trials evaluated the benefits of dual antiplatelet therapy and the associated bleeding risk in stroke prevention. The Management of Atherothrombosis With Clopidogrel in High-

Risk Patients trial reported that aspirin plus clopidogrel did not reduce the risk of major events but increased major bleeding risk compared with clopidogrel alone [16]. In addition, the Fast Assessment of Stroke and Transient Ischemic Attack to Prevent Early Recurrence (FASTER) trial found that aspirin plus clopidogrel did not significantly decrease stroke risk at 90 days and did not increase the hemorrhagic rate compared with aspirin alone [17]. However, several other studies reported that clopidogrel plus aspirin reduced the risk of recurrent stroke compared with aspirin alone [18, 19].

Previous studies found that Asian patients were at a higher risk for cerebrovascular disease. The carrier rate of the CYP2C19 loss-of-function variant was higher in Asian populations, which might affect the efficacy of clopidogrel [20, 21]. Aspirin's failure in preventing recurrent stroke is not uncommon, and clopidogrel may not be the most suitable antiplatelet drug for the Chinese population. Given that specific antiplatelet treatments that might be more efficient in preventing recurrent stroke in Asian patients have not been investigated to date, we conducted a population-based case-cohort study to compare the efficacy of aspirin, clopidogrel, and ticlopidine in preventing recurrent ischemic stroke in Taiwanese patients.

## Methods

### Data source and ethics approval

This population-based case-cohort study used data from the Taiwan National Health Insurance Research Database (NHIRD) that comprises data obtained from millions of people. The National Health Insurance (NHI) program in Taiwan has operated since 1995. The NHIRD is a research database developed by the NHI Research Institute and contains patient healthcare data from hospitals, outpatient clinics, and community pharmacies. It encompasses more than 99% of 23 million individuals and 95% of the hospitals in Taiwan. The NHI Research Institute provides the database to researchers after anonymizing all personal information. The current study includes data retrieved from the "Longitudinal Health Insurance Database" (LHID 2005) from a random sample of one million individuals within the NHIRD, with linked longitudinal data available from 2000 to 2010. The LHID 2005 contains complete medical claims and registration for a random sample of one million individuals within the NHIRD. The randomized data (LHID 2005) are represent all beneficiaries as there are no significant differences in sex, age, and premium rate between individuals in the LHID 2005 and the original NHIRD data sets. The codes of the Internal Classification of Disease, Nine Revision (ICD-9) were used to define diseases. This study was approved by the Institutional Review Board of the Ditmanson Medical Foundation Chiayi Christian Hospital, Taiwan (CYCH-IRB: 2018013).

### Study subjects and definitions

Ischemic stroke was defined as an episode of the neurological deficit by cerebral infarction confirmed by imaging (computed tomography or magnetic resonance imaging) [22]. All patients hospitalized with a primary diagnosis of ischemic stroke (ICD-9 codes, 433, 434, and 436) from January 1, 2000, to December 31, 2010, were included in the study. The exclusion criteria were the following: age younger than 18 years; a history of stroke before January 1, 2000; a history of myocardial infarction, atrial fibrillation, or infective endocarditis; use of antiplatelet drugs for more than one month before the first stroke; recurrent stroke within one month after the first stroke; use of more than one antiplatelet drug. We believe that the patients who did not regularly use antiplatelet drugs potentially had poor medical compliance. To avoid bias, we excluded patients who did not use antiplatelet drugs for more than 14 days within the first month after the stroke and patients who had not used antiplatelet drugs for more than 90 consecutive days.

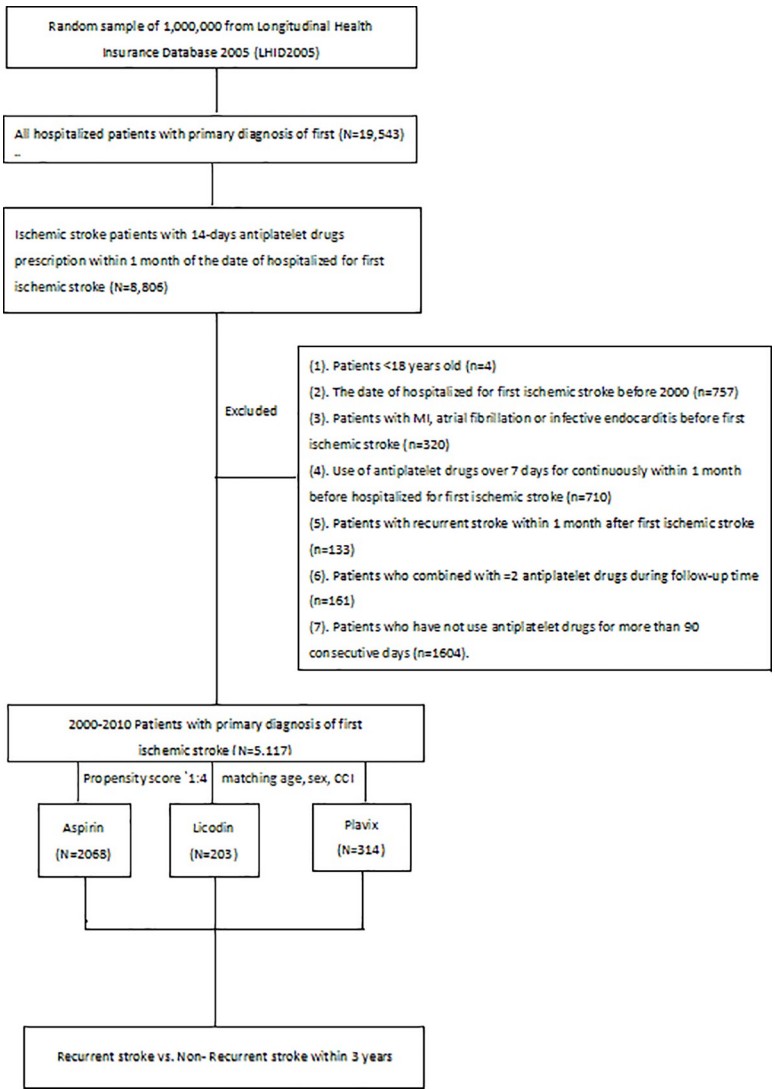

**Fig 1. Flow chart of patient enrollment.** CCI: clinical comorbidity index, Licodin: ticlopidine, Plavix: clopidogrel.

A total of 19,543 patients were hospitalized with the discharge diagnosis of ischemic stroke. Among these, 8806 patients, who used antiplatelet drugs for more than 14 days during the first month after stroke onset, were included in the study (Fig 1). The patients were categorized into three groups: aspirin (aspirin, 100 mg/day), ticlopidine (ticlopidine, 250 mg/day), and clopidogrel (clopidogrel, 75 mg/day). Age, sex, and clinical comorbidity index (CCI) were used as propensity score matching parameters. All patients who used ticlopidine or clopidogrel for stroke prevention were included in the study. Two separate propensity score matches were performed using a ratio of 1:4 to compare the aspirin group with the ticlopidine and the clopidogrel groups.

## Outcome measures

The primary endpoint of the study was the development of a new ischemic or hemorrhagic stroke. Patients who did not use antiplatelet drugs for more than 90 days were defined as those not on antiplatelet drugs and were not included in the analyses. The patients who changed

antiplatelet drugs or died due to causes not related to stroke were not included in the study. The endpoint was the clinical condition of patients on regular antiplatelet treatment who did not experience a stroke for a minimum of three years.

## Statistical analysis

In the current study, we used propensity score matching with a ratio of 1:4 to match the study patients based on age, sex, and CCI. The baseline characteristics of the patients in categorical and continuous variables were compared using the chi-square test. Because the prescription of antiplatelet agents for stroke prevention changed over time, a time-dependent Cox proportional hazards model was used to compare the risk of recurrent stroke between the propensity score-matched groups. Hazard ratios were used to measure the related risk of recurrent stroke. Because of the high mortality rate in stroke patients, we performed competing-risk regression by the Fine and Gray model. We also used stratified analysis to determine the risk of recurrent stroke among patients according to diabetes mellitus and peptic ulcer status. The Kaplan-Meier method was used to analyze the proportion of patients with recurrent stroke during the follow-up period. All statistical analyses were performed using SAS version 9.4 (SAS Institute, Inc, Cary, NC). Two-sided P-values of $< .05$ were considered statistically significant.

## Results

From 2000 to 2010, a total of 5117 patients fulfilled the inclusion criteria for the current study. After the 1:4 propensity score matching for age, sex, and CCI, a total of 2585 patients who were included in the analysis, including 2068, 203, and 314 patients in the aspirin, ticlopidine, and clopidogrel groups, respectively. The patients' baseline characteristics, including sex, age, hypertension status, stroke severity, and hemorrhagic stroke history, were not significantly different among the three groups. The length of hospital stay was longer in the clopidogrel group; the rates of diabetes mellitus and hyperlipidemia were lower in the ticlopidine group. The rate of peptic ulcers was higher in the clopidogrel group. The rate of thrombocytopenia as an adverse effect was not significantly different among the three groups (Table 1).

During the 3-year follow-up period, the recurrent ischemic stroke rates were 2.03% (42/2068), 1.48% (3/203), and 2.55% (8/314) in the aspirin, ticlopidine, and clopidogrel groups, respectively. The time-dependent Cox proportional hazards model analysis determined that age, CCI, clopidogrel use, diabetes mellitus, peptic ulcers, thrombocytopenia, and hemorrhagic stroke history were associated with an increased risk of recurrent stroke. After adjusting for the risk factors that affect recurrent stroke, age, CCI, clopidogrel use, and history of hemorrhage were associated with an increased risk of recurrent stroke (Table 2). The competing risk-adjusted hazard ratio (HR) of recurrent stroke was higher in patients with a hemorrhagic stroke history and those on clopidogrel treatment for stroke prevention (2.79 [1.02–7.61] and 2.27 [1.02–5.07], respectively). Compared with aspirin, ticlopidine did not increase the risk of recurrent stroke (competing risk-adjusted HR, 0.62 [0.08–4.80]) (Table 2). The Kaplan-Meier curves indicated that the time to recurrent stroke (Fig 2) was significantly different among the three groups (P < .0001, log-rank test).

The hemorrhagic stroke rates were not significantly different among the three groups (p = .85), 1.40% (29/2068), 1.48% (3/203), and 0.95% (3/314) in the aspirin, ticlopidine, and clopidogrel groups, respectively. Neutropenia was found in 0.29% (6/2068) of patients receiving aspirin, whereas no neutropenia was found in patients receiving ticlopidine and clopidogrel.

In patients with diabetes mellitus, age was associated with an increased risk of recurrent stroke, whereas the risk of recurrent stroke was not significantly different among those treated with aspirin, clopidogrel, and ticlopidine (Table 3). Among patients with peptic ulcers, age and

**Table 1. The characteristics of the stroke patients at baseline.**

| Variables | Aspirin | Ticlopidine | Clopidogrel | P value |
|---|---|---|---|---|
| Number | 2068 | 203 | 314 | |
| Age, years | | | | |
| 20–44 | 63 (3.05%) | 4 (1.97%) | 14 (4.46%) | 0.481 |
| 45–59 | 485 (23.45%) | 53 (26.11%) | 76 (24.2%) | |
| ≥60 | 1520 (73.5%) | 146 (71.92%) | 224 (71.34%) | |
| Mean ± SD | 67.66 ± 11.59 | 68.28 ± 11.64 | 67.22 ± 12.05 | 0.602 |
| Sex | | | | |
| Female | 799 (38.64%) | 87 (42.86%) | 114 (36.31%) | 0.326 |
| Male | 1269 (61.36%) | 116 (57.14%) | 200 (63.69%) | |
| Length of stay, days | | | | |
| <9 | 1684 (81.43%) | 170 (83.74%) | 230 (73.25%) | 0.002 |
| ≥9 | 384 (18.57%) | 33 (16.26%) | 84 (26.75%) | |
| CCI, mean ± SD | 4.46 ± 2.58 | 4.25 ± 2.43 | 4.65 ± 2.71 | 0.218 |
| Diabetes mellitus | 1046 (50.58%) | 84 (41.38%) | 153 (48.73%) | 0.041 |
| Hypertension | 1819 (87.96%) | 176 (86.70%) | 275 (87.58%) | 0.864 |
| Hyperlipidemia | 1166 (56.38%) | 96 (47.29%) | 183 (58.28%) | 0.030 |
| CKD | 155 (7.50%) | 13 (6.40%) | 37 (11.78%) | 0.023 |
| Peptic ulcer disease | 865 (41.83%) | 91 (44.83%) | 212 (67.52%) | <0.001 |
| Thrombocytopenia | 16 (0.77%) | 3 (1.48%) | 6 (1.91%) | 0.084 |
| Hemorrhagic stroke | 53 (2.56%) | 7 (3.45%) | 11 (3.5%) | 0.52 |

CCI, clinical comorbidity index; CKD, chronic kidney disease; SD, standard deviation

hemorrhagic stroke history were associated with an increased risk of recurrent stroke (HR, 8.18 [2.11–31.61]). In contrast, recurrent stroke risk was not significantly different among the aspirin, clopidogrel, and ticlopidine groups (Table 4).

## Discussion

The present population-based case-cohort study included the data of 1,000,000 randomly selected individuals in Taiwan and revealed three major findings. First, clopidogrel was associated with a higher recurrent stroke rate than aspirin among patients on antiplatelet treatment for secondary stroke prevention. Second, the risk of recurrent stroke was higher in patients with ischemic stroke and hemorrhagic stroke history. Third, among patients with diabetes mellitus and gastric ulcers, the recurrent stroke risk with aspirin treatment was not significantly different from that with clopidogrel or ticlopidine treatment.

Our finding that the recurrent stroke rate was not significantly different between the ticlopidine and aspirin groups agrees with the results reported by Gorelick et al., who showed that the two-year recurrent stroke rates with ticlopidine and aspirin were 11.9% and 9.5%, respectively (P = 0.1) [8]. The current study's recurrent stroke rate is lower than those reported in previous studies [8, 23], which might be partly because of the exclusion of patients who died due to causes other than stroke and those who experienced recurrent stroke within one month after the first stroke. The lower competing risk-adjusted HR of recurrent stroke in the ticlopidine group (0.62 [0.08–4.86]) compared with the aspirin group in the current study is comparable to that reported by Hass et al., who showed that ticlopidine was slightly more effective than aspirin in preventing stroke [23].

**Table 2. The relationship of antiplatelet drug use with 3-year recurrent stroke among patients with first-time ischemic stroke using the time-dependent Cox proportional hazards model.**

| Variables | Crude HR (95%CI) | Adjusted HR (95%CI) | Competing risk-adjusted HR (95%CI) |
|---|---|---|---|
| Age, year | 1.05 (1.02–1.08) | 1.05 (1.02–1.09) | 1.03 (0.99–1.06) |
| Sex | | | |
| Female | Ref. | Ref. | Ref. |
| Male | 1.25 (0.72–2.18) | 1.58 (0.86–2.91) | 1.64 (0.91–2.94) |
| Length of stay, days | | | |
| <9 | Ref. | Ref. | Ref. |
| ≥9 | 1.03 (0.54–1.98) | 1.01 (0.52–1.99) | 0.96 (0.50–1.82) |
| CCI | 1.23 (1.12–1.35) | 1.18 (1.03–1.35) | 1.11 (0.99–1.23) |
| Antiplatelet agents | | | |
| Aspirin | Ref. | Ref. | Ref. |
| Ticlopidine | 0.69 (0.10–5.07) | 0.71 (0.10–5.29) | 0.62 (0.08–4.86) |
| Clopidogrel | 2.64 (1.34–5.22) | 2.16 (1.01–4.65) | 2.27 (1.02–5.07) |
| Diabetes mellitus | | | |
| No | Ref. | Ref. | Ref. |
| Yes | 1.89 (1.08–3.30) | 1.21 (0.65–2.28) | 1.28 (0.67–2.46) |
| Hypertension | | | |
| No | Ref. | Ref. | Ref. |
| Yes | 3.58 (0.49–25.91) | 2.45 (0.33–18.11) | 2.75 (0.37–20.6) |
| Hyperlipidemia | | | |
| No | Ref. | Ref. | Ref. |
| Yes | 0.84 (0.49–1.46) | 0.83 (0.47–1.46) | 0.84 (0.49–1.46) |
| CKD | | | |
| No | Ref. | Ref. | Ref. |
| Yes | 1.83 (0.65–5.10) | 0.92 (0.30–2.87) | 0.93 (0.30–2.91) |
| Peptic ulcer disease | | | |
| No | Ref. | Ref. | Ref. |
| Yes | 2.08 (1.20–3.58) | 1.23 (0.66–2.32) | 1.36 (0.70–2.65) |
| Thrombocytopenia | | | |
| No | Ref. | Ref. | Ref. |
| Yes | 6.88 (2.13–22.23) | 3.16 (0.85–11.76) | 3.31 (0.69–15.94) |
| Hemorrhagic stroke | | | |
| No | Ref. | Ref. | Ref. |
| Yes | 4.38 (1.57–12.2) | 3.10 (1.05–9.18) | 2.79 (1.02–7.61) |

*Time-dependent Cox proportional hazards model, **ICU was not adjusted in time-dependent Cox proportional hazard model; CCI, clinical comorbidity index; CI, confidence interval

CKD, chronic kidney disease; HR, hazard ratio

In the CAPRIE trial, a randomized, blinded study evaluating the relative efficacy of clopidogrel (75 mg once daily) and aspirin (325 mg once daily), clopidogrel was more effective than aspirin in the prevention of stroke, myocardial infarction, and death related to vascular disease [13]. However, another study reported no significant difference in the efficacy of aspirin and clopidogrel for the prevention of recurrent stroke or functional outcomes [24]. The present study results revealed that the recurrent stroke risk was higher in the clopidogrel group than in the aspirin group. This finding agrees with the studies by Liu et al. and Wang et al., who showed that the carrier rate of the cytochrome P450 2C19 loss-of-function variant was higher in Asian populations, which may affect clopidogrel's efficacy [20, 21]. However, we did not

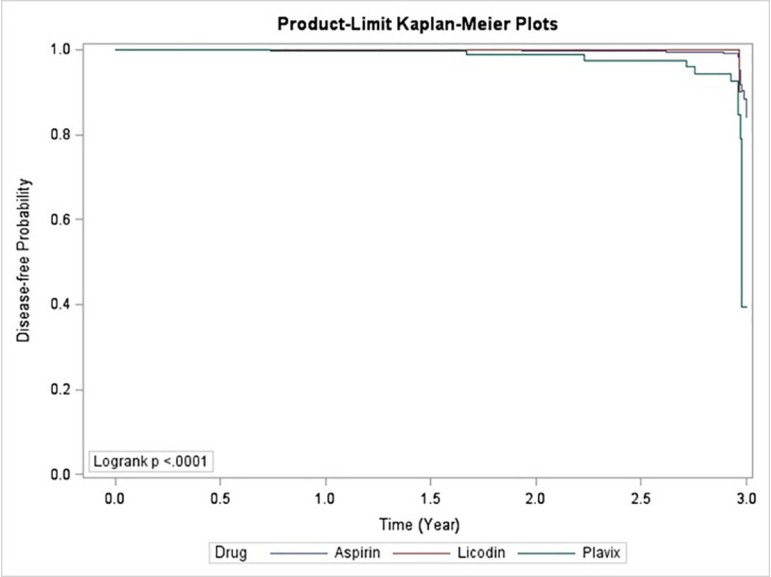

**Fig 2. Kalpan–Meier curve for recurrent stroke.** Licodin: ticlopidine, Plavix: clopidogrel.

find a significant difference in the recurrent stroke risk between the aspirin and clopidogrel groups among patients with diabetes mellitus or those with gastric ulcers. This difference between the current study and previous reports might be related to the NHI guidelines. Specifically, clopidogrel can only be used in patients with gastric ulcers and those with a contraindication for aspirin. Patients without gastric ulcers who are prescribed clopidogrel might harbor other risk factors that may affect the risk of recurrent stroke. However, whether gastric or upper gastrointestinal bleeding might affect the risk of recurrent stroke remains unclear.

Compared with the current study, aspirin and ticlopidine doses were significantly higher in the previous studies investigating secondary stroke prevention [8, 13, 23, 25]. At present, the commonly used daily doses of aspirin, ticlopidine, and clopidogrel for stroke prevention in Taiwan are 100, 250, and 75 mg, respectively. Clopidogrel and ticlopidine have similar structures, and both are metabolized by the hepatic cytochrome P450 1A enzyme to acquire activity. Since the active metabolite is the same for both compounds, in theory, the dose for stroke prevention should be the same for both compounds. However, in previous studies, the doses of ticlopidine and clopidogrel were 500 and 75 mg/day, respectively, with ticlopidine being more than six times that of clopidogrel. Since 75 mg/day clopidogrel was confirmed to be as effective as aspirin, 75 mg/day ticlopidine may provide the effect achieved by 100 mg/day aspirin or 75 mg clopidogrel. In the current study, 250 mg/day ticlopidine was not inferior to 100 mg/day aspirin in preventing recurrent stroke, and 75 mg/day clopidogrel was associated with a higher risk of recurrent stroke. These results are comparable to those reported by Uchiyama et al., who found that the effect of 200 mg/day ticlopidine was comparable to 75 mg/day clopidogrel for the secondary prevention of vascular events in patients with ischemic stroke [26].

Hematologic abnormalities, including neutropenia and thrombocytopenia, due to ticlopidine, were reported. In studies in which patients received ticlopidine 500 mg/day, the incidence of neutropenia was between 0.6% and 3.4% [8, 23, 27, 28]. The incidence of neutropenia in patients who used ticlopidine 250 mg/day was between 0.29% and 0.37% [29, 30]. The incidence of neutropenia seems related to the dose of this antiplatelet.

The incidence of neutropenia in patients who used aspirin 650 mg/day is 2.2% [8]. A lower ticlopidine dose can reduce the risk of neutropenia. In the study, neutropenia was not found

**Table 3. The relationship of antiplatelet drug use and 3-year recurrent stroke among patients with first-time ischemic stroke using the time-dependent Cox proportional hazards model stratified by diabetes mellitus.**

| | Diabetes mellitus | |
|---|---|---|
| | No | Yes |
| Variables | Adjusted HR (95%CI) | Adjusted HR (95%CI) |
| Age, year | 1.04 (0.99–1.09) | 1.07 (1.02–1.12) |
| Sex | | |
| Female | Ref. | Ref. |
| Male | 1.29 (0.44–3.80) | 1.73 (0.77–3.86) |
| Length of stay, days | | |
| <9 | Ref. | Ref. |
| ≥9 | 0.23 (0.04–1.20) | 1.64 (0.71–3.77) |
| CCI | 1.21 (0.98–1.49) | 1.19 (0.97–1.46) |
| Antiplatelet agents | | |
| Aspirin | Ref. | Ref. |
| Ticlopidine | <0.01 (<0.01–NA) | 1.66 (0.21–13.28) |
| Clopidogrel | 1.12 (0.27–4.62) | 2.62 (0.99–6.95) |
| Hypertension | | |
| No | Ref. | Ref. |
| Yes | 1.01 (0.13–8.06) | >999.99 (<0.01–NA) |
| Hyperlipidemia | | |
| No | Ref. | Ref. |
| Yes | 1.32 (0.47–3.74) | 0.76 (0.36–1.61) |
| CKD | | |
| No | Ref. | Ref. |
| Yes | 11.47 (2.68–49.2) | 0.21 (0.03–1.69) |
| Peptic ulcer disease | | |
| No | Ref. | Ref. |
| Yes | 2.77 (0.92–8.37) | 1.14 (0.50–2.63) |
| Thrombocytopenia | | |
| No | Ref. | Ref. |
| Yes | 28.88 (3.17–263.24) | 1.72 (0.19–15.20) |
| Hemorrhagic stroke | | |
| No | Ref. | Ref. |
| Yes | 2.51 (0.29–22.07) | 3.18 (0.86–11.82) |

*Time-dependent Cox proportional hazard model

CCI, clinical comorbidity index; CI, confidence interval; CKD, chronic kidney disease

HR, hazard ratio

in patients receiving ticlopidine and clopidogrel. The result is uncertain about our patients receiving a low dose of ticlopidine (250 mg/day). Our study found that the risk of intracranial hemorrhage is not significantly different among patients receiving aspirin, ticlopidine, and clopidogrel. These results are similar to those reported by previous studies that the hemorrhagic stroke rate was not significantly different between aspirin and clopidogrel [1, 13] and between aspirin and ticlopidine [23].

Overall, these results suggest that a ticlopidine dose of 500 mg/day may not be necessary for the secondary prevention of recurrent stroke. The adverse event of severe neutropenia might be related to the high ticlopidine dose. Our study showed that 250 mg/day ticlopidine was

**Table 4. The relationship of antiplatelet drug use and 3-year recurrent stroke among patients with first-time ischemic stroke using the time-dependent Cox proportional hazards model stratified by peptic ulcer disease.**

| | Peptic ulcer disease | |
|---|---|---|
| | No | Yes |
| | Adjusted HR (95%CI) | Adjusted HR (95%CI) |
| Age, year | 1.05 (1.00–1.11) | 1.04 (1.00–1.09) |
| Sex | | |
| Female | Ref. | Ref. |
| Male | 1.48 (0.56–3.91) | 1.78 (0.77–4.15) |
| Length of stay, days | | |
| <9 | Ref. | Ref. |
| ≥9 | 0.91 (0.28–2.93) | 1.29 (0.54–3.09) |
| CCI | 1.40 (1.11–1.77) | 1.10 (0.92–1.31) |
| Antiplatelet agents | | |
| Aspirin | Ref. | Ref. |
| Ticlopidine | <0.01 (<0.01–NA) | 1.33 (0.17–10.49) |
| Clopidogrel | 3.70 (0.81–16.99) | 1.65 (0.66–4.16) |
| Diabetes mellitus | | |
| No | Ref. | Ref. |
| Yes | 1.12 (0.42–2.97) | 1.10 (0.45–2.73) |
| Hypertension | | |
| No | Ref. | Ref. |
| Yes | >999.99 (<0.01–NA) | 1.25 (0.16–9.69) |
| Hyperlipidemia | | |
| No | Ref. | Ref. |
| Yes | 0.77 (0.30–1.96) | 0.84 (0.38–1.86) |
| CKD | | |
| No | Ref. | Ref. |
| Yes | 1.51 (0.28–8.15) | 0.80 (0.16–3.91) |
| Thrombocytopenia | | |
| No | Ref. | Ref. |
| Yes | 5.46 (0.38–78.26) | 3.36 (0.65–17.38) |
| Hemorrhagic stroke | | |
| No | Ref. | Ref. |
| Yes | 0.71 (0.08–6.68) | 8.18 (2.11–31.62) |

*Time-dependent Cox proportional hazards model

**ICU was not adjusted in time-dependent Cox proportional hazard model

CCI, clinical comorbidity index; CI, confidence interval; CKD, chronic kidney disease

HR, hazard ratio

superior to 75 mg/day clopidogrel for recurrent stroke prevention. The antithrombotic effect of clopidogrel is influenced by the patient's CYP2C19 genotype. Because of CYP2C19 genetic polymorphism, the response of clopidogrel differs widely among patients. Previous studies showed that adjusting thienopyridine treatment in patients after primary percutaneous coronary intervention for ST-Elevation Myocardial Infarction, according to the CYP2C19 genotype, can improve a patient's outcome [31, 32]. Whether adjusting the clopidogrel dose after CYP2C19 genotyping in stroke patients can reduce recurrent stroke risk needs further investigation.

## Limitations

The current study has several limitations. First, data on complete blood and platelet counts were not available in the NHIRD, which did not permit the analyses of thrombocytopenia and neutropenia. Second, the information on certain potential confounding factors, such as smoking, alcohol use, body mass index, and low-density lipoprotein cholesterol, were not available in the NHIRD. This inaccessibility may have confounded the association between antiplatelet drugs and recurrent stroke risk. Third, the recurrent stroke definition in the current study was based on the diagnosis of ischemic stroke among hospitalized patients, which might have underestimated the recurrent stroke rate. Fourth, the data were over 10 years old; the actual values of these findings for current stroke treatment remain uncertain. Fifth, we excluded patients who did not regularly use antiplatelet drugs, which may exclude patients who discontinued antiplatelet medications due to an adverse effect. These exclusions may underestimate the risk of adverse effects of antiplatelet medications. However, in Taiwan, when adverse effects occur, most patients will come to the hospital for help, and physicians will change the drug but will not discontinue medications.

## Conclusion

Compared with patients using aspirin, those using clopidogrel were at a higher risk of recurrent stroke. For secondary stroke prevention, aspirin was superior to clopidogrel, whereas ticlopidine was not inferior to aspirin.

## Supporting information

**S1 Table. Condition of antiplatelet use.**
(DOCX)

## Acknowledgments

We thank Chih-Cheng Hsu from the National Health Research Institute Taiwan, for providing valuable comments on data collection and statistical analysis.

## Author Contributions

**Conceptualization:** Yueh-Han Hsu, Cheung-Ter Ong.

**Data curation:** Ching-Fang Tsai.

**Formal analysis:** Ching-Fang Tsai.

**Investigation:** Yueh-Han Hsu.

**Methodology:** Yi-Sin Wong, Yueh-Han Hsu.

**Project administration:** Yueh-Han Hsu, Cheung-Ter Ong.

**Software:** Ching-Fang Tsai.

**Supervision:** Cheung-Ter Ong.

**Writing – original draft:** Yi-Sin Wong.

**Writing – review & editing:** Cheung-Ter Ong.

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
