## [Decision Letter · Decision Letter 0]

4 Aug 2020

PONE-D-20-18884

The efficacy of aspirin, clopidogrel, and ticlopidine in stroke prevention: a population-based case-cohort study in Taiwan

PLOS ONE

Dear Dr. Ong,

Thank you for submitting your manuscript to PLOS ONE. After careful consideration, we feel that it has merit but does not fully meet PLOS ONE’s publication criteria as it currently stands. Therefore, we invite you to submit a revised version of the manuscript that addresses the points raised during the review process.

Please see my comments below. 

We look forward to receiving your revised manuscript.

Kind regards,

Hugo ten Cate, MD, PhD

Academic Editor

PLOS ONE

Additional Editor Comments:

Two issues may be added to the discussion section. One is a limitation, indicating that the data were over 10 years old, hence the actual value of these findings for current stroke treatment remain uncertain. A second issue that could be added is that instead of considering higher doses of clopidogrel genotyping of patients for the relevant polymorphisms may be a more appealing approach to verify metabolism of clopidogrel. This approach may be quite helpful, asa discussed in the recent commentary by ten Berg et al, JACC Cardiovascular Intervention 2020, at least for the coronary population. Evidence for stroke patient still needs to be established.

Journal Requirements:

2. In the ethics statement in the manuscript and in the online submission form, please provide additional information about the patient records used in your retrospective study. Specifically, please ensure that you have discussed whether all data were fully anonymized before you accessed them and/or whether the IRB or ethics committee waived the requirement for informed consent. If patients provided informed written consent to have data from their medical records used in research, please include this information. In addition, please include the dates upon which this data was accessed.

Reviewers' comments:

Reviewer's Responses to Questions

**Comments to the Author**

1. Is the manuscript technically sound, and do the data support the conclusions?

Reviewer #1: Yes

Reviewer #2: Yes

2. Has the statistical analysis been performed appropriately and rigorously? 

Reviewer #1: Yes

Reviewer #2: Yes

3. Have the authors made all data underlying the findings in their manuscript fully available?

Reviewer #1: Yes

Reviewer #2: Yes

4. Is the manuscript presented in an intelligible fashion and written in standard English?

Reviewer #1: Yes

Reviewer #2: Yes

5. Review Comments to the Author

Reviewer #1: Methods state "The primary endpoint of the study was the development of a new ischemic or

hemorrhagic stroke." It would be helpful for the authors to break down this outcome by ischemic v hemorrhagic, as in some studies of antiplatelet medications alone and in combination, the benefit for ischemic stroke prevention is outweighed by increased risk of hemorrhage. Therefore the authors should report the effect of these 3 antiplatelet agents on both recurrent ischemic stroke and new hemorrhagic stroke.

Reviewer #2: Summary of the research

The present manuscript by Wong et al reports the results of a case-cohort study comparing recurrent stroke rates among Taiwanese patients discharged with ischemic stroke and treated with either aspirin, ticlopidine or clopidogrel monotherapy. A random sample of 1-million patients was extracted from a National Health Insurance Research database. Patients (n=19543) hospitalized between January 2000 and December 2010 with a primary diagnosis of ischemic stroke were identified and after propensity score matching 3102 patients were left to be included in this analysis. The authors found that for recurrent stroke prevention, aspirin was superior to clopidogrel and ticlopidine non-inferior to aspirin.

Overall impression

The manuscript gives a clear introduction addressing the key problems and knowledge gaps in stroke prevention among the Asian population and the importance of this topic. However, there are issues that need to be addressed.

• The methods of data collection should be more detailed. How was the 1-million sample ‘randomly’ collected? Why chosen for 1-million sample instead of directly identifying patients from the NHIRD based on discharge diagnosis in a certain time span?

• Of the patients identified with primary diagnosis ischemic stroke (n=19543) >50% were excluded. As this is a large number excluded from the population of interest (in a cohort study), I would like to gain more insight in the exclusion criteria.

• To be able to interpret the role of different antithrombotic agents, not only thrombotic risks should be reported, but also bleeding risks. Is there any information available on bleeding complications among patients on aspirin vs ticlopidine vs clopidogrel therapy?

• Ticlopidine was found to be non-inferior to aspirin. However, ticlopidine use is world-wide mainly limited due to its serious side effect such as neutropenia and TTP. Although data on neutropenia and thrombocytopenia were not available in the NHIRD, I believe the prevalence of these ticlopidine side effects among Asian population should be highlighted more in the discussion to put results in better perspective.

Specific comments:

• The manuscript reports that, based on the time-dependent Cox proportional hazards model analysis, length of stay was associated with increased risk of recurrent stroke. However, the crude HR for length of stay >9d was 1.03 (0.54-1.98).

• The manuscript reports that after adjusting for risk factors, length of stay and thrombocytopenia were associated with increased risk of recurrent stroke. However, the adjusted HR for length of stay >9d was 1.01 (0.52-1.99) and for thrombocytopenia was 3.16 (0.85-11.76).

• Add reference in the last paragraph of the Discussion stating: “…Altogether, these results suggest that a ticlopidine dose of 500mg/day may not be necessary for the prevention of recurrent stroke and that the adverse event of severe neutropenia might be related to the high ticlopidine dose.”

6. PLOS authors have the option to publish the peer review history of their article (what does this mean?). If published, this will include your full peer review and any attached files.

Reviewer #1: No

Reviewer #2: No

---

## [Author Response · Author response to Decision Letter 0]

1 Sep 2020

Response to editor

Thank editor important comments. Base on editor comments we have made necessary revision. In the comments, two issue was pointed out. Now we present our revision.

1. One is a limitation, indicating that the data were over 10 years old, hence the actual value of these findings for current stroke treatment remain uncertain.

Revision: In the last sentence of limitation section, we add the sentence “Fourth, the data were over 10 years old, the actual value of these finding for current stroke treatment remain uncertain.” 

2. A second issue that could be added is that instead of considering higher doses of clopidogrel genotyping of patients for the relevant polymorphisms may be a more appealing approach to verify metabolism of clopidogrel.

Revision: In discussion section last sentence, we add the revision. “The antithrombotic effect of clopidogrel is influenced by the patient’s CYP2C19 genotype. Because of CYP2C19 genetic polymorphism, the response of clopidogrel differ widely among patients. Previous studies showed according CYP2C19 genotype adjust thienopyridine treatment in patients after primary PCI for STEMI can improve patient’s outcome. Whether adjust clopidogrel dose after CYP2C19 genotyping in stroke patients can reduce risk of recurrent stroke need further investigation.”

Response to reviewer 1

We thank reviewer valuable comments. Base on reviewer’s comments, we have made necessary revision.

Comments: The primary endpoint of the study was the development of a new ischemic or hemorrhagic stroke." It would be helpful for the authors to break down this outcome by ischemic v hemorrhagic, as in some studies of antiplatelet medications alone and in combination, the benefit for ischemic stroke prevention is outweighed by increased risk of hemorrhage. Therefore the authors should report the effect of these 3 antiplatelet agents on both recurrent ischemic stroke and new hemorrhagic stroke.

Revision:

1. In the result section, we add the information of neutropenia and hemorrhagic stroke. “The hemorrhagic stroke rates were no significant difference among 3 groups (p=0.85), 1.40% (29/2068), 1.48% (3/203), and 0.95% (3/314) in the aspirin, ticlopidine, and clopidogrel groups, respectively. Neutropenia was found in 0.29% (6/2068) of patients receiving aspirin, no neutropenia was found in patients receiving ticlopidine and clopidogrel.”

2. In discussion section, we add the discussion about neutropenia and hemorrhage stroke.

“Hematologic abnormalities including neutropenia and thrombocytopenia attribute to ticlopidine had been reported. In the studies which patient receiving ticlopidine 500mg/day, the incidence of neutropenia was between 0.6% and 3.4%. The incidence of neutropenia in patients receiving ticlopidine 250mg/day was between 0.29% and 0.37%. The incidence of neutropenia seems related to the dose of antiplatelet.

The incidence of neutropenia in the patients use aspirin 650 mg/day is 2.2%. Lower ticlopidine dose can reduce the risk of neutropenia. In the study, neutropenia was not found in patients receiving ticlopidine and clopidogrel. The result suspect related to our patients receiving low dose of ticlopidine (250mg/day). Our study found the risk of intracranial hemorrhage is no significant difference among the patients receiving aspirin, ticlopidine and clopidogrel. The result is the same as previous studies that the hemorrhage stroke rate was no significant difference between aspirin and clopidogrel and between aspirin and ticlopidine.

Response to reviewer 2

We thank reviewer's valuable comments. Base on reviewer’s comments, we have made necessary revision.

1. The methods of data collection should be more detailed. How was the 1-million sample ‘randomly’ collected? Why chosen for 1-million sample instead of directly identifying patients from the NHIRD based on discharge diagnosis in a certain time span?

Revision: In the methods section, we have add the information about the methods of data collection. “The current study included the data were retrieved from the “Longitudinal Health Insurance Database (LHID 2005) from a random sample of 1 million individuals within the NHIRD, with linked longitudinal data available from 2000 to 2010. The LHID 2005 contains complete medical claims and registration for a random sample of one million individuals within the NHIRD. The randomized data (LHID 2005) are overall representative of all beneficiaries as no significant difference in the sex, age and premium rate between individuals in the LHID 2005 and in the original NHIRD data sets.”

2. Of the patients identified with primary diagnosis ischemic stroke (n=19543) >50% were excluded. As this is a large number excluded from the population of interest (in a cohort study), I would like to gain more insight in the exclusion criteria.

Revision: In the methods section, we add the exclusion criteria. “We excluded the patients who did not use of one of the three antiplatelet drugs for more than 14 days within the first month after the stroke and the patients not continued use of the antiplatelet drugs were not included in the study.”

3. To be able to interpret the role of different antithrombotic agents, not only thrombotic risks should be reported, but also bleeding risks. Is there any information available on bleeding complications among patients on aspirin vs ticlopidine vs clopidogrel therapy? 

Revision: In the results and discussion section, we add the information about hemorrhage stroke and neutropenia.

 In results section: “The hemorrhagic stroke rates were no significant difference among 3 groups (p=0.85), 1.40% (29/2068), 1.48% (3/203), and 0.95% (3/314) in the aspirin, ticlopidine, and clopidogrel groups, respectively. Neutropenia was found in 0.29% (6/2068) of patients receiving aspirin, no neutropenia was found in patients receiving ticlopidine and clopidogrel.”

 In discussion section: “Hematologic abnormalities including neutropenia and thrombocytopenia attribute to ticlopidine had been reported. In the studies which patient use ticlopidine 500mg/day, the incidence of neutropenia was between 0.6% and 3.4%. The incidence of neutropenia in patients use ticlopidine 250mg/day was between 0.29% and 0.37%.31, 32 The incidence of neutropenia seems related to the dose of antiplatelet.

The incidence of neutropenia in the patients use aspirin 650 mg/day is 2.2%. Lower ticlopidine dose can reduce the risk of neutropenia. In the study, neutropenia was not found in patients receiving ticlopidine and clopidogrel. The result suspect related to our patients receiving low dose of ticlopidine (250mg/day). Our study found the risk of intracranial hemorrhage is no significant difference among the patients receiving aspirin, ticlopidine and clopidogrel. The result is the same as previous studies that the hemorrhage stroke rate was no significant difference between aspirin and clopidogrel and between aspirin and ticlopidine.

4. Ticlopidine was found to be non-inferior to aspirin. However, ticlopidine use is world-wide mainly limited due to its serious side effect such as neutropenia and TTP. Although data on neutropenia and thrombocytopenia were not available in the NHIRD, I believe the prevalence of these ticlopidine side effects among Asian population should be highlighted more in the discussion to put results in better perspective.

Revision: Neutropenia and hemorrhage adverse effect was reported and discussion as issue 3.

5. • The manuscript reports that, based on the time-dependent Cox proportional hazards model analysis, length of stay was associated with increased risk of recurrent stroke. However, the crude HR for length of stay >9d was 1.03 (0.54-1.98).

Revision: We have deleted length of stay.

6. The manuscript reports that after adjusting for risk factors, length of stay and thrombocytopenia were associated with increased risk of recurrent stroke. However, the adjusted HR for length of stay >9d was 1.01 (0.52-1.99) and for thrombocytopenia was 3.16 (0.85-11.76).

Revision: We have deleted length of stay and thrombocytopenia.

7. Add reference in the last paragraph of the Discussion stating: “…Altogether, these results suggest that a ticlopidine dose of 500mg/day may not be necessary for the prevention of recurrent stroke and that the adverse event of severe neutropenia might be related to the high ticlopidine dose.”

Revision: In discussion section, we have add the reference about the relationship between ticlopidine dose and neutropenia or thrombocytopenia. “Hematologic abnormalities including neutropenia and thrombocytopenia attribute to ticlopidine had been reported. In the studies which patient receiving ticlopidine 500mg/day, the incidence of neutropenia was between 0.6% and 3.4%.27-30. The incidence of neutropenia in patients use ticlopidine 250mg/day was between 0.29% and 0.37%.31, 32 The incidence of neutropenia seems related to the dose of antiplatelet.

The incidence of neutropenia in the patients use aspirin 650 mg/day is 2.2%.30 Lower ticlopidine dose can reduce the risk of neutropenia. In the study, neutropenia was not found in patients receiving ticlopidine and clopidogrel. The result suspect related to our patients receiving low dose of ticlopidine (250mg/day). Our study found the risk of intracranial hemorrhage is no significant difference among the patients receiving aspirin, ticlopidine and clopidogrel. The result is the same as previous studies that the hemorrhage stroke rate was no significant difference between aspirin and clopidogrel and between aspirin and ticlopidine.

---

## [Decision Letter · Decision Letter 1]

25 Sep 2020

PONE-D-20-18884R1

The efficacy of aspirin, clopidogrel, and ticlopidine in stroke prevention: a population-based case-cohort study in Taiwan

PLOS ONE

Dear Dr. Ong,

Thank you for submitting your manuscript to PLOS ONE. After careful consideration, we feel that it has merit but does not fully meet PLOS ONE’s publication criteria as it currently stands. Therefore, we invite you to submit a revised version of the manuscript that addresses the points raised during the review process.

We look forward to receiving your revised manuscript.

Kind regards,

Hugo ten Cate, MD, PhD

Academic Editor

PLOS ONE

Additional Editor Comments (if provided):

One reviewer has some additional remarks that merit your attention.

Reviewers' comments:

Reviewer's Responses to Questions

**Comments to the Author**

1. If the authors have adequately addressed your comments raised in a previous round of review and you feel that this manuscript is now acceptable for publication, you may indicate that here to bypass the “Comments to the Author” section, enter your conflict of interest statement in the “Confidential to Editor” section, and submit your "Accept" recommendation.

Reviewer #2: (No Response)

2. Is the manuscript technically sound, and do the data support the conclusions?

Reviewer #2: Partly

3. Has the statistical analysis been performed appropriately and rigorously? 

Reviewer #2: Yes

4. Have the authors made all data underlying the findings in their manuscript fully available?

Reviewer #2: Yes

5. Is the manuscript presented in an intelligible fashion and written in standard English?

Reviewer #2: Yes

6. Review Comments to the Author

Reviewer #2: To the authors,

Thank you for your responses and hard work. I would like to address two more issues:

1. Of patients identified with primary diagnosis ischemic stroke (n=19543) >50% were excluded. The authors report in the revised version exclusion of the patients who did not use one of the three antiplatelet drugs for more than 14 days within the first month after the stroke and the patients not continued use of the antiplatelet drugs. I recommend the authors to explain this latter exclusion criteria, be more specific: discontinuation after what period of treatment. Discontinuation of antiplatelet drugs during treatment can also be due to bleeding or ischemic complications. In that case exclusion of these patients resulted in selection bias with possible influence of study results.

2. Figure 1 reports 8,806 patients screened for inclusion. Based on the exclusion criteria, 8806-2085=6721 patients should be included for propensity score matching (2085 = sum of criteria 1-6 in figure). Author report 5117, thus 6721 – 5117 = 1604 patients were excluded currently for unknown reason. Please correct the numbers or report the reason of exclusion of these 1604 patients.

7. PLOS authors have the option to publish the peer review history of their article (what does this mean?). If published, this will include your full peer review and any attached files.

Reviewer #2: No

---

## [Author Response · Author response to Decision Letter 1]

29 Sep 2020

Response to reviewer 2

Thanks reviewer’s comments, based on reviewer’s comments we have made necessary change.

1. Of patients identified with primary diagnosis ischemic stroke (n=19543) >50% were excluded. The authors report in the revised version exclusion of the patients who did not use one of the three antiplatelet drugs for more than 14 days within the first month after the stroke and the patients not continued use of the antiplatelet drugs. I recommend the authors to explain this latter exclusion criteria, be more specific: discontinuation after what period of treatment. Discontinuation of antiplatelet drugs during treatment can also be due to bleeding or ischemic complications. In that case exclusion of these patients resulted in selection bias with possible influence of study results. 

Response: In methods “Study subjects and definitions” section, Page 8 to page 9, we add a sentence “We believe the patients who did not regularly use antiplatelet drugs potentially poor medical compliance. To reduce bias, we excluded the patients who did not use antiplatelet drugs for more than 14 days within the first month after the stroke and the patients who had not use antiplatelet drugs for more than 90 consecutive days .”

 In figure 1, we add an exclusion criteria, “(7). Patients who have not use antiplatelet drugs for more than 90 consecutive days (n=1604).”

2. Figure 1 reports 8,806 patients screened for inclusion. Based on the exclusion criteria, 8806-2085=6721 patients should be included for propensity score matching (2085 = sum of criteria 1-6 in figure). Author report 5117, thus 6721 – 5117 = 1604 patients were excluded currently for unknown reason. Please correct the numbers or report the reason of exclusion of these 1604 patients.

Response: In figure 1. We add an exclusion criteria. “(7). Patients who had not use antiplatelet drugs for more than 90 consecutive days (n=1604).”

Response to editor

We thank reviewer valuable comments. Base on reviewer’s comments, we have made necessary change.

1. In the patient exclusion criteria, we have add a sentence “We believe the patients who did not regularly use antiplatelet drugs potentially poor medical compliance. To reduce bias, we excluded the patients who did not use antiplatelet drugs for more than 14 days within the first month after the stroke and the patients who had not use antiplatelet drugs for more than 90 consecutive days . 

 2. In figure 1, we add an exclusion criteria. “Patients who had not use antiplatelet drugs for more than 90 consecutive days (n=1604).

---

## [Decision Letter · Decision Letter 2]

22 Oct 2020

PONE-D-20-18884R2

The efficacy of aspirin, clopidogrel, and ticlopidine in stroke prevention: a population-based case-cohort study in Taiwan

PLOS ONE

Dear Dr. Ong,

Thank you for submitting your manuscript to PLOS ONE. After careful consideration, we feel that it has merit but does not fully meet PLOS ONE’s publication criteria as it currently stands. Therefore, we invite you to submit a revised version of the manuscript that addresses the points raised during the review process.

We look forward to receiving your revised manuscript.

Kind regards,

Hugo ten Cate, MD, PhD

Academic Editor

PLOS ONE

Additional Editor Comments (if provided):

One reviewer still wants you to address the possibility of selection bias and I think the question is reasonable. Could you take one moor look?

Reviewers' comments:

Reviewer's Responses to Questions

**Comments to the Author**

1. If the authors have adequately addressed your comments raised in a previous round of review and you feel that this manuscript is now acceptable for publication, you may indicate that here to bypass the “Comments to the Author” section, enter your conflict of interest statement in the “Confidential to Editor” section, and submit your "Accept" recommendation.

Reviewer #2: (No Response)

2. Is the manuscript technically sound, and do the data support the conclusions?

Reviewer #2: Partly

3. Has the statistical analysis been performed appropriately and rigorously? 

Reviewer #2: Yes

4. Have the authors made all data underlying the findings in their manuscript fully available?

Reviewer #2: Yes

5. Is the manuscript presented in an intelligible fashion and written in standard English?

Reviewer #2: No

6. Review Comments to the Author

Reviewer #2: To the authors,

Thank you for considering my comments. However, I still have concern regarding possible selection bias in this study.

You describe that patients did not use APT for more than >90 consecutive days due to poor medical compliance and therefore you excluded these patients form this study to reduce bias. However, it is not known whether this hypothesis is correct. It is likely that patients did not use APT for >90 days due to bleeding complications. And if poor medical compliance played a role, it could also be the result of nuisance bleeding. I would have suggested to include also patients not that did not use APT for more than >90 consecutive days in the primary analysis and afterwards, exclude this group in a sensitivity analysis.

However, as the study is now, I believe that there is a possible relevant selection bias which at least should be mentioned as a limitation in the discussion.

7. PLOS authors have the option to publish the peer review history of their article (what does this mean?). If published, this will include your full peer review and any attached files.

Reviewer #2: No

---

## [Author Response · Author response to Decision Letter 2]

30 Oct 2020

We thanks reviewer’s valuable comments. We very concern selection bias, because it may influence the result, so we excluded the patients who potential poor medical compliance. However, it may also excluded the patients who discontinue antiplatelet due to major adverse effect of antiplatelet. It is the limitation of the retrospective study from the data base. We add a paragraph in limitation section. Two issue was point out by reviewer, we have made necessary revision. 

1. Is the manuscript presented in an intelligible fashion and written in standard English?

Response: The English have been proofreading and editing by Enago-Taiwan 

(enago.tw). 

2. Thank you for considering my comments. However, I still have concern regarding possible selection bias in this study.

You describe that patients did not use APT for more than >90 consecutive days due to poor medical compliance and therefore you excluded these patients form this study to reduce bias. However, it is not known whether this hypothesis is correct. It is likely that patients did not use APT for >90 days due to bleeding complications. And if poor medical compliance played a role, it could also be the result of nuisance bleeding. I would have suggested to include also patients not that did not use APT for more than >90 consecutive days in the primary analysis and afterwards, exclude this group in a sensitivity analysis.

However, as the study is now, I believe that there is a possible relevant selection bias which at least should be mentioned as a limitation in the discussion.

Response: 

In the limitation section, we add a limitation “Fifth, we excluded patients who did not regularly use antiplatelet drugs, which may also exclude patients who discontinued antiplatelet medications due to an adverse effect. These exclusions may underestimate the risk of adverse effects of antiplatelet medications. However, in Taiwan, when adverse effects occur, most patients will come to the hospital for help, and physicians will change the drug but will not discontinue medications.”

---

## [Editor Report · Decision Letter 3]

3 Nov 2020

Efficacy of Aspirin, Clopidogrel, and Ticlopidine in Stroke Prevention: A Population-Based Case-Cohort Study in Taiwan

PONE-D-20-18884R3

Dear Dr. Ong,

We’re pleased to inform you that your manuscript has been judged scientifically suitable for publication and will be formally accepted for publication once it meets all outstanding technical requirements.

Kind regards,

Hugo ten Cate, MD, PhD

Academic Editor

PLOS ONE
---

## [Editor Report · Acceptance letter]

8 Dec 2020

PONE-D-20-18884R3 

Efficacy of Aspirin, Clopidogrel, and Ticlopidine in Stroke Prevention: A Population-Based Case-Cohort Study in Taiwan 

Dear Dr. Ong:

I'm pleased to inform you that your manuscript has been deemed suitable for publication in PLOS ONE. Congratulations! Your manuscript is now with our production department. 

Kind regards, 

on behalf of

Prof. Hugo ten Cate 

Academic Editor

PLOS ONE